# Sustainable and Healthy Eating Behaviors and Environmental Literacy of Generations X, Y and Z with the Same Ancestral Background: A Descriptive Cross-Sectional Study

Neslihan Öner [1], Hasan Durmuş [2], Yağmur Yaşar Fırat [1], Arda Borlu [2,*] and Nilüfer Özkan [3]

1. Department of Nutrition and Dietetics, Faculty of Health Sciences, Erciyes University, 38280 Kayseri, Türkiye; neslihancelik@erciyes.edu.tr (N.Ö.); yagmuryasar@erciyes.edu.tr (Y.Y.F.)
2. Department of Public Health, Faculty of Medicine, Erciyes University, 38280 Kayseri, Türkiye; hasandurmus@erciyes.edu.tr
3. Department of Nutrition and Dietetics, Faculty of Health Sciences, Ordu University, 52200 Ordu, Türkiye; niluferozkan@odu.edu.tr
* Correspondence: ardab@erciyes.edu.tr

**Abstract:** As social culture and structure evolve, changes occur in individuals' eating habits and environmental awareness. This study assesses the relationship between sustainable eating behaviors and environmental literacy across generations (Gens) from the same ancestry. The Sustainable and Healthy Eating (SHE) Behavior Scale and the Environmental Literacy Scale for Adults (ELSA) was administered to 381 individuals across three generations. Self-reported anthropometric data were collected. The total scores of the SHE Behavior Scale of the participants from all three generations were significantly different from each other. The "Quality Labels", "Meat Reduction" and "Low Fat" factor scores were similar in GenX and GenY. These factor scores were significantly lower in GenZ. The "Animal Welfare" factor score was significantly higher in GenX. The "Avoiding Food Waste and Seasonal Foods" and "Local Food" factor scores were significantly higher in GenX than in GenY and GenZ. The "Healthy and Balanced Diet" factor score was significantly lower in GenZ. There was no difference between the total ELSA scores. The "Environmental Consciousness" factor score was significantly lower in GenX than in GenY. Generational disparities strongly influence perspectives on sustainable and healthy eating. Focused initiatives are essential to educate future parents, who play a pivotal role in shaping the next generation, about sustainable nutrition.

**Keywords:** ancestry; environmental literacy; generations; nutrition; sustainable eating behaviors

## 1. Introduction

Throughout the historical evolution of humanity, there has been a consistent tendency for individuals to collaborate, aiming to ensure survival and foster development. Thus, it becomes inevitable that individuals living in the same period and growing up under comparable conditions are shaped within a similar sociocultural environment [1]. The collective experiences of individuals exposed to analogous historical and social contexts have played a pivotal role in influencing their lifestyles and dietary behaviors across their lifespans, contributing to the emergence of distinct dietary habits among various generations [1–3]. In a study conducted by Zeren et al. [4], it was observed that individuals who are members of different generations have different nutritional behaviors; Generations X and Y prefer a healthier nutritional perspective; however, Generation Z exhibits a hedonistic approach to eating habits. Although the nutritional knowledge of Generation Z is higher than other generations, it is stated that their interest in nutrition is more related to their physical appearance. Different perceptions about nutrition are likely to create different eating habits in different generations.

Eating habits are one of the important factors affecting an individual's lifestyle and health [5]. These eating habits are shaped by the social, cultural, and economic connections

of the period in which the individual grows up. In this context, the fact that the nutritional characteristics of GenX, GenY and GenZ are different from each other is associated with being influenced by different historical and social contexts [6]. GenX is a generation that witnessed the industrial era and grew up with traditional eating habits. GenX, which generally focuses on basic food groups and prefers foods obtained from natural sources, strictly adheres to traditional norms on nutrition [7,8]. GenY is a generation that has grown up in a period of the rapid evolution of technology and has kept pace with this change. With the rise of fast-food culture, millennials tend to eat faster and more practically, while at the same time being conscious about healthy eating. GenZ, on the other hand, is a generation that is more likely to be more conscious about environmental health [9] and sustainable nutrition in an age where digitalization has reached its peak. Sustainability-oriented trends such as plant-based nutrition, organic product preference and the zero-waste movement, and fast-food nutrition, a product of popular culture, are among the important factors shaping the dietary preferences of GenZ [6].

Rapid population growth and unsustainable eating behaviors, coupled with the pressures of climate change, threaten the food security of future generations and are critical issues that need to be urgently addressed [10]. It is very important that sustainable diets, defined by the Food and Agriculture Organization (FAO) as "nutritionally adequate, safe and healthy diets that meet the nutritional needs of present and future generations, respect biodiversity and ecosystems, are protective, culturally acceptable, accessible, economically affordable, nutritionally adequate, safe and healthy", are adopted by members of society from all generations [11,12].

Environmental literacy is the ability to perceive and interpret the health of environmental systems and to adopt appropriate behaviors to improve, renew and maintain the health of these systems [11]. The existence of individuals with high levels of environmental literacy is very important for the prevention of environmental problems that are experienced in today's world and that are likely to be experienced in the future, thus ensuring sustainability [11]. In order to increase public awareness and consciousness of the impact of diet on climate change and the environment, it is possible that a multidimensional approach, including the socio-demographic characteristics of individuals, will be beneficial [13]. From this perspective, the combination of sustainable eating behaviors and environmental awareness and consciousness across generations is a topic worth examining. To the best of our knowledge, there is no study that evaluates sustainable and healthy eating behaviors and environmental literacy together across generational differences. The aim of this study is to assess the relationship between sustainable and healthy eating behaviors and environmental literacy of members of GenX, GenY and GenZ with the same ancestral background.

The main research questions were as follows:

*RQ1* Is there a relationship between sustainable and healthy eating behaviors and the environmental literacy of different generations?

*RQ2* Which factors affect the sustainable and healthy eating behaviors and environmental literacy of different generations?

## 2. Materials and Methods

Data for this descriptive and cross-sectional study were collected between December 2022 and February 2023. All members of GenX, GenY and GenZ with the same ancestral background who agreed to participate were included in the study. The same ancestral background of the participants was determined as grandparent–parent–grandchild. Families with foreign nationals and families with GenZ members under the age of 15 were excluded. The sample size was calculated with G*Power program. With the Many Groups ANOVA test, at least 323 people were included in the calculation, made over 3 groups with a medium effect size: α: 0.05, β: 95%, df: 5. Considering the possibility of erroneous and incomplete information, it was aimed to include 130 families in the study to reach 20% more participants than the calculated minimum sample size. One hundred-and-thirty-three

families were reached through social media invitations. In two of these families, a family member in GenZ and a family member in GenX did not agree to participate in the study, and the study was completed with a total of 127 families and 381 people. The actual power calculated by post hoc power analysis was 97%. The participants were invited to the study via social media platforms. Data of the study were collected using the Google Forms platform. Considering that there may be regional or cultural differences in the classification of generations in this study, the age ranges of a study conducted in Turkey [6] were preferred; individuals representing GenX were defined as those born between 1965 and 1979, individuals representing GenY were defined as those born between 1980 and 1999, and individuals representing GenZ were defined as those born in 2000 and later.

Data collection tools including socio-demographic characteristics, anthropometric measurements, the Sustainable and Healthy Eating (SHE) Behaviors Questionnaire and the Environmental Literacy Scale (ELSA) was used. Height and weight were recorded according to self-reporting, and body mass index (BMI) was calculated. The Turkish validity and reliability study of the SHE Behaviors Questionnaire, developed by Żakowska-Biemans [14], was conducted by Köksal et al. [15]. The 7 factors on this are "Healthy and Balanced Nutrition", "Quality Labels", "Meat Reduction", "Local Food", "Low Fat", "Avoiding Food Waste and Seasonal Foods" and "Animal Welfare". The 32 items in the scale are evaluated on a 7-point Likert scale and each item is labeled as "Never", "Very rarely", "Rarely", "Sometimes", "Often", "Very often" or "Always". "Never = 1" and "Always = 7" points were accepted. Compliance with SHE behaviors increases as the score obtained from the scale increases. The factor scores of the subscales are calculated by averaging the scores given to the items in that factor (min 1–max 7). The total scale score is calculated by mean of all factor scores. An increase in the total score obtained from the scale is associated with more SHE behaviors.

The Environmental Literacy Scale for Adults (ELSA), developed by Atabek-Yiğit et al. [10], was used to determine the level of environmental literacy of participants. This scale consists of a total of 20 items in three factors: "Environmental Consciousness", "Environmental Anxiety" and "Environmental Awareness". All items in the scale consist of 5-point Likert-type statements corresponding to values defined as 'Strongly agree = 5', 'Agree = 4', 'Undecided = 3', 'Disagree = 2' and 'Strongly disagree = 1'. The lowest score that can be obtained from the scale is 20 and the highest score is 100. A high score on the ELSA indicates that the individual has a high level of environmental literacy, while a low score indicates a low level of environmental literacy. According to the total score obtained from ELSA, scores between 20 and 36 are classified as 'Very low', scores between 37 and 52 as 'Low', scores between 53 and 68 as 'Moderate', scores between 69 and 84 as 'High' and scores between 85 and 100 as 'Very high'.

The study procedure was performed according to Declaration of Helsinki and STROBE checklist [16]. The study was approved by the Social and Human Sciences Ethics Committee of Erciyes University (Date: 14 October 2022 and Approval No: 2022/04.08). Participants were informed about the study and their written consent was obtained. Written consent was also obtained from underage participants' parents, who were also participants in the same study, who were informed and gave consent for their children's participation.

Data analysis was performed using the IBM SPSS Statistics (Statistical Package for the Social Sciences, SPSS Inc., Chicago, IL, USA) 22.0 statistical package program. The Shapiro–Wilk test was used to assess the normality of the data. The mean and standard deviation ($\overline{X} \pm$ SD) were reported for normally distributed data, and median and mode were reported for non-normally distributed data. Descriptive information for categorical data was presented as numbers (n) and percentages (%). One-way ANOVA test was used to compare normally distributed data and the Kruskal–Wallis test was used for non-normally distributed data. Tukey's post hoc test was used to determine the group from which the statistical difference originated. A chi-square test was used to analyze categorical data. Correlation of data was assessed by Pearson correlation analysis. $p < 0.05$ was considered significant.

## 3. Results

In this study, which involved 381 participants sharing the same ancestral background and comprising members of GenX, GenY, and GenZ, the mean age of GenX participants was $51.72 \pm 4.56$ years, the mean age of GenY members was $31.58 \pm 7.57$ years, and the mean age of GenZ members was $18.96 \pm 2.94$ years. The number of main meals consumed daily by GenY, GenY and GenZ were, respectively, $2.43 \pm 0.58$, $2.26 \pm 0.63$, and $2.42 \pm 0.63$ ($p > 0.05$). While there was no difference between the generations in terms of main meal consumption, the number of snacks consumed daily by GenZ participants was found to be statistically significantly higher than the number of snacks consumed by GenX participants (GenX = $1.21 \pm 0.88$, GenY = $1.35 \pm 0.86$, and GenZ = $1.54 \pm 0.83$, $p = 0.011$). It was found that 65.8% of GenZ participants skipped breakfast and the most common reason for skipping meals among GenZ participants was 'not being able to wake up in the morning' at 31.5%. Sociodemographic characteristics and some eating behaviors of GenX, GenY and GenZ participants were shown in Table 1.

**Table 1.** Sociodemographic characteristics and some eating behaviors of GenX, GenY and GenZ participants.

| Characteristics | GenX (n = 127) | | GenY (n = 127) | | GenZ (n = 127) | | *p* |
|---|---|---|---|---|---|---|---|
| | n | % | n | % | n | % | |
| **Gender** | | | | | | | |
| Man | 23 | 18.1 | 25 | 19.7 | 40 | 31.5 | **0.022** |
| Woman | 104 | 81.9 | 102 | 80.3 | 87 | 68.5 | |
| **Marital status** | | | | | | | |
| Married | 107 | 84.3 | 60 | 47.2 | 8 | 6.3 | |
| Single | 8 | 6.3 | 64 | 50.4 | 119 | 93.7 | **<0.001** |
| Other | 12 | 9.4 | 3 | 2.4 | 0 | 0.0 | |
| **Where most of life is spent** | | | | | | | |
| Province | 69 | 54.3 | 83 | 65.3 | 82 | 64.6 | |
| District | 35 | 27.6 | 33 | 26.0 | 32 | 25.2 | 0.161 |
| Village | 23 | 18.1 | 11 | 8.7 | 13 | 10.2 | |
| **Education status** | | | | | | | |
| Illiterate | 15 | 11.8 | 4 | 3.1 | 0 | 0.0 | |
| Primary school | 38 | 29.9 | 14 | 11.1 | 3 | 2.3 | |
| Secondary school | 23 | 18.1 | 6 | 4.7 | 17 | 13.4 | **<0.001** |
| High school | 25 | 19.7 | 23 | 18.1 | 43 | 33.9 | |
| University | 21 | 16.6 | 73 | 57.5 | 64 | 50.4 | |
| Postgraduate | 5 | 3.9 | 7 | 5.5 | 0 | 0.0 | |
| **Employment status** | | | | | | | |
| Working | 40 | 31.5 | 58 | 45.7 | 11 | 8.7 | **<0.001** |
| Not working | 87 | 68.5 | 69 | 54.3 | 116 | 91.3 | |

**Table 1.** *Cont.*

| Characteristics | GenX (n = 127) | | GenY (n = 127) | | GenZ (n = 127) | | *p* |
|---|---|---|---|---|---|---|---|
| | n | % | n | % | n | % | |
| **Sector of employment \*** | | | | | | | |
| Informatics | 0 | 0.0 | 3 | 5.2 | 0 | 0.0 | |
| Education | 8 | 20.0 | 14 | 24.1 | 1 | 9.1 | |
| Finance | 2 | 5.0 | 5 | 8.6 | 0 | 0.0 | |
| Health | 6 | 15.0 | 10 | 17.2 | 4 | 36.4 | **<0.001** |
| Marketing | 2 | 5.0 | 6 | 10.3 | 2 | 18.1 | |
| Tourism | 0 | 0.0 | 4 | 6.9 | 0 | 0.0 | |
| Other | 22 | 55.0 | 16 | 27.7 | 4 | 36.4 | |
| **Income status** | | | | | | | |
| Income less than expenditure | 34 | 26.7 | 38 | 29.9 | 40 | 31.5 | |
| Income equals expenditure | 58 | 45.7 | 55 | 43.3 | 59 | 46.5 | 0.868 |
| Income more than expenditure | 35 | 27.6 | 34 | 26.8 | 28 | 22.0 | |
| **Membership to environmental organization** | | | | | | | |
| Yes | 8 | 6.3 | 12 | 9.4 | 19 | 15.0 | 0.105 |
| No | 119 | 93.7 | 115 | 90.6 | 108 | 85.0 | |
| **Parenting status** | | | | | | | |
| Yes | 123 | 96.9 | 59 | 46.5 | 2 | 1.6 | **<0.001** |
| No | 4 | 3.1 | 68 | 53.5 | 125 | 98.4 | |
| **Mother's education level** | | | | | | | |
| Illiterate | 55 | 43.3 | 11 | 8.7 | 0 | 0.0 | |
| Primary school | 62 | 48.8 | 55 | 43.3 | 39 | 30.7 | |
| Secondary school | 8 | 6.3 | 22 | 17.3 | 17 | 13.4 | **<0.001** |
| High school | 1 | 0.8 | 27 | 21.2 | 42 | 33.1 | |
| University | 1 | 0.8 | 12 | 9.5 | 29 | 22.8 | |
| **Father's education level** | | | | | | | |
| Illiterate | 23 | 18.1 | 3 | 2.4 | 0 | 0.0 | |
| Primary school | 61 | 48.1 | 29 | 22.8 | 22 | 17.3 | |
| Secondary school | 13 | 10.2 | 29 | 22.8 | 26 | 20.5 | **<0.001** |
| High school | 21 | 16.5 | 32 | 25.2 | 34 | 26.8 | |
| University | 9 | 7.1 | 34 | 26.8 | 45 | 35.4 | |
| **Cigarette use** | | | | | | | |
| Yes | 24 | 18.9 | 26 | 20.5 | 25 | 19.7 | 0.951 |
| No | 103 | 81.1 | 101 | 79.5 | 102 | 80.3 | |
| **Alcohol use** | | | | | | | |
| Yes | 2 | 1.6 | 8 | 6.3 | 7 | 5.5 | 0.374 |
| No | 125 | 97.4 | 119 | 93.7 | 120 | 94.5 | |

**Table 1.** *Cont.*

| Characteristics | GenX (n = 127) | | GenY (n = 127) | | GenZ (n = 127) | | *p* |
|---|---|---|---|---|---|---|---|
| | n | % | n | % | n | % | |
| **Skipping main meals** | | | | | | | |
| Yes | 61 | 48.0 | 72 | 56.7 | 82 | 64.6 | 0.101 |
| No | 66 | 52.0 | 55 | 43.3 | 45 | 35.4 | |
| Skipped meal | | | | | | | |
| Breakfast | 19 | 31.5 | 37 | 51.3 | 54 | 65.8 | |
| Lunch | 39 | 64.1 | 32 | 44.5 | 27 | 32.9 | <0.001 |
| Dinner | 3 | 4.4 | 3 | 4.2 | 1 | 1.3 | |
| **Chronic disease status** | | | | | | | |
| Yes | 61 | 48.0 | 13 | 10.2 | 6 | 4.7 | <0.001 |
| No | 66 | 52.0 | 114 | 89.8 | 121 | 95.3 | |
| **Regular medication use** | | | | | | | |
| Yes | 59 | 46.5 | 17 | 13.4 | 6 | 4.7 | <0.001 |
| No | 68 | 53.5 | 110 | 86.6 | 121 | 95.3 | |
| **Use of vitamin–mineral supplements** | | | | | | | |
| Yes | 40 | 31.5 | 54 | 42.5 | 45 | 35.4 | 0.258 |
| No | 87 | 68.5 | 73 | 57.5 | 82 | 64.6 | |

Note: Descriptive statistics of normally distributed data were given as $\overline{X} \pm SD$. An independent two-sample *t*-test was used to analyze the data. * Only participants who declared that they were employed.

Anthropometric measurements of the GenX, GenY and GenZ participants are presented in Table 2. Body weight and BMI median values of the participants from all three generations were significantly different from each other. The median value of the height of GenX participants was significantly lower than the median value of the height of GenY and GenZ participants ($p < 0.001$). It was determined that 29.9% of GenX participants, 12.6% of GenY participants and 5.5% of GenZ participants were obese. The rates of being overweight and obese were significantly higher in GenX participants than in GenY and GenZ participants ($p < 0.001$).

**Table 2.** Anthropometric measurements of the GenX, GenY and GenZ participants.

| Anthropometric Measurements | GenX (n = 127) | | GenY (n = 127) | | GenZ (n = 127) | | *p* |
|---|---|---|---|---|---|---|---|
| | Median (Min–Max) | | Median (Min–Max) | | Median (Min–Max) | | |
| **Weight (kg)** | 70 (50–110) [a] | | 65 (45–113) [b] | | 60 (30–105) [c] | | <0.001 |
| **Height (cm)** | 160 (145–195) [a] | | 165 (151–198) [b] | | 166 (134–195) [b] | | <0.001 |
| **BMI (kg\m$^2$)** | 27.7 (19.84–41.67) [a] | | 23.8 (17.0–42.53) [b] | | 21.7 (15.59–34.1) [c] | | <0.001 |
| **BMI classification** | n | % | n | % | n | % | *p* |
| Underweight | 0 | 0.0 | 7 | 5.5 | 15 | 11.8 | |
| Normal | 26 | 20.5 | 68 | 53.5 | 86 | 67.7 | <0.001 |
| Overweight | 63 | 49.6 | 36 | 28.4 | 19 | 15.0 | |
| Obese | 38 | 29.9 | 16 | 12.6 | 7 | 5.5 | |

Note: Descriptive statistics of categorical data are given as n and %. Chi-square test was used to analyze the data. Superscript letters indicate statistical difference between groups ($p < 0.05$).

Total and factor scores of the SHE Behaviors Scale and ELSA for GenX, GenY and GenZ participants were shown in Table 3. The total score of the SHE Behaviors Scale of the participants from all three generations were significantly different from each other ($p < 0.001$). The "Quality Labels" ($p = 0.001$), "Meat Reduction" ($p = 0.011$) and "Low Fat" ($p = 0.026$) factor scores were similar in GenX and GenY. These factor scores were significantly lower in GenZ. The "Animal Welfare" factor score was significantly higher in GenX ($p = 0.006$). The "Avoiding Food Waste and Seasonal Foods" ($p < 0.001$) and "Local Food" ($p < 0.001$) factor scores were significantly higher in GenX participants than in GenY and GenZ participants. The "Healthy and Balanced Nutrition" factor score was similar in GenX and GenY participants, whereas it was significantly lower in GenZ participants ($p < 0.001$). There was no difference between the total score obtained from the ELSA for GenX, GenY and GenZ participants. However, the "Environmental Consciousness" factor score was significantly lower in GenX participants than in GenY participants ($p = 0.007$).

**Table 3.** Total and factor scores of the SHE Behaviors Scale and ELSA for GenX, GenY and GenZ participants.

| Scores | GenX (n = 127) Median (Min–Max) | | GenY (n = 127) Median (Min–Max) | | GenZ (n = 127) Median (Min–Max) | | *p* |
|---|---|---|---|---|---|---|---|
| **SHE Behaviors Scale** | | | | | | | |
| Quality Labels * | 4.33 ± 0.94 [a] | | 4.18 ± 0.98 [a] | | 3.89 ± 1.00 [b] | | **0.001** |
| Avoiding Food Waste and Seasonal Foods | 5.14 (2.43–7) [a] | | 4.57 (1.86–7) [b] | | 4.43 (1–7) [b] | | **<0.001** |
| Animal Welfare | 4.5 (1.5–7) [a] | | 4 (1–7) [b] | | 3.75 (1–7) [b] | | **0.006** |
| Meat Reduction | 4 (1–7) [a] | | 3.67 (1–7) [a] | | 3.33 (1–7) [b] | | **0.011** |
| Healthy and Balanced Nutrition | 5 (1.75–7) [a] | | 5 (2.25–7) [a] | | 4.5 (1–7) [b] | | **<0.001** |
| Local Food | 4 (1–7) [a] | | 3 (1–7) [b] | | 3 (1–7) [b] | | **<0.001** |
| Low Fat | 5 (2–7) [a] | | 5 (2.33–7) [a] | | 4.67 (1–7) [b] | | **0.026** |
| **Total *** | 4.56 ± 0.78 [a] | | 4.27 ± 0.83 [b] | | 4.02 ± 0.93 [c] | | **<0.001** |
| **ELSA scores** | | | | | | | |
| Environmental Consciousness | 39 (25–45) [a] | | 41 (17–45) [b] | | 41 (24–45) [a,b] | | **0.007** |
| Environmental Anxiety | 26 (14–30) | | 27 (12–30) | | 27 (10–30) | | 0.310 |
| Environmental Awareness | 19 (10–25) | | 19 (9–25) | | 19 (5–25) | | 0.879 |
| **Total** | 84 (51–100) | | 87 (44–100) | | 87 (46–100) | | 0.149 |
| **ELSA classification ** ** | | | | | | | |
| Non-high | 16 | 12.6 | 5 | 3.9 | 11 | 8.7 | **0.045** |
| High | 111 | 87.4 | 122 | 96.1 | 116 | 91.3 | |

Note: * Descriptive statistics of normally distributed data are given as $\overline{X} \pm$ SD. Independent-two sample *t*-test was used to analyze the data. ** ELSA score classification was analyzed, with those whose scale score was "Very low", "Low" and "Moderate" represented as "Non-high" and those whose scale score was "High" and "Very high" represented as "High". Descriptive statistics of categorical data are given as n and %. Chi-square test was used to analyze the data. Superscript letters indicate statistical difference between groups ($p < 0.05$).

The correlation of SHE Behaviors and ELSA scores within GenX, GenY, and GenZ participants is shown in Table 4. SHE behaviors were found to be associated with the generation Z, while there was no significant relationship between GenX and GenZ. However, there was significant positive correlation in the change in environmental literacy in all three generations.

**Table 4.** The correlation of SHE Behaviors and ELSA scores within GenX, GenY, and GenZ participants.

| Scores | GenX (n = 127) | GenY (n = 127) | GenZ (n = 127) |
|---|---|---|---|
| **Total SHE Behavior Score** | | | |
| GenX | 1 | | |
| GenY | **0.233 \*\*** | 1 | |
| GenZ | 0.62 | **0.221 \*** | 1 |
| **Quality Labels** | | | |
| GenX | 1 | | |
| GenY | **0.197 \*** | 1 | |
| GenZ | **0.232 \*\*** | **0.251 \*** | 1 |
| **Avoiding Food Waste and Seasonal Foods** | | | |
| GenX | 1 | | |
| GenY | −0.009 | 1 | |
| GenZ | 0.004 | 0.134 | 1 |
| **Animal Welfare** | | | |
| GenX | 1 | | |
| GenY | **0.363 \*\*** | 1 | |
| GenZ | 0.118 | **0.269 \*\*** | 1 |
| **Meat Reduction** | | | |
| GenX | 1 | | |
| GenY | **0.353 \*\*** | 1 | |
| GenZ | 0.278 | **0.376 \*\*** | 1 |
| **Healthy and Balanced Nutrition** | | | |
| GenX | 1 | | |
| GenY | **0.177 \*** | 1 | |
| GenZ | **0.174 \*** | **0.229 \*\*** | 1 |
| **Local Food** | | | |
| GenX | 1 | | |
| GenY | **0.321 \*\*** | 1 | |
| GenZ | **0.279 \*\*** | **0.211 \*** | 1 |
| **Low Fat** | | | |
| GenX | 1 | | |
| GenY | 0.128 | 1 | |
| GenZ | −0.023 | 0.107 | 1 |
| **ELSA Total** | | | |
| GenX | 1 | | |
| GenY | **0.402 \*\*** | 1 | |
| GenZ | **0.396 \*\*** | **0.417 \*\*** | 1 |
| **Environmental Consciousness** | | | |
| GenY | 1 | | |
| GenZ | **0.410 \*\*** | 11 | |
| GenY | **0.369 \*\*** | **0.362 \*\*** | 1 |

**Table 4.** *Cont.*

| Scores | GenX (n = 127) | GenY (n = 127) | GenZ (n = 127) |
|---|---|---|---|
| **Environmental Anxiety** | | | |
| GenY | 1 | | |
| GenZ | 0.330 ** | 1 | |
| GenY | 0.326 ** | 0.396 ** | |
| **Environmental Awareness** | | | |
| GenY | 1 | | |
| GenZ | 0.280 ** | 1 | |
| GenY | 0.335 ** | 0.330 ** | 1 |

\* $p < 0.05$, \*\* $p < 0.01$.

The correlation between the scores on the SHE Behaviors Scale and the ELSA of the GenX, GenY, and GenZ participants are shown in Figure 1. A significant, positive, and weak correlation was found between the scores of the SHE Behaviors Scale and the ELSA of participants of GenX ($p = 0.048$), GenY ($p = 0.002$) and GenZ ($p < 0.001$).

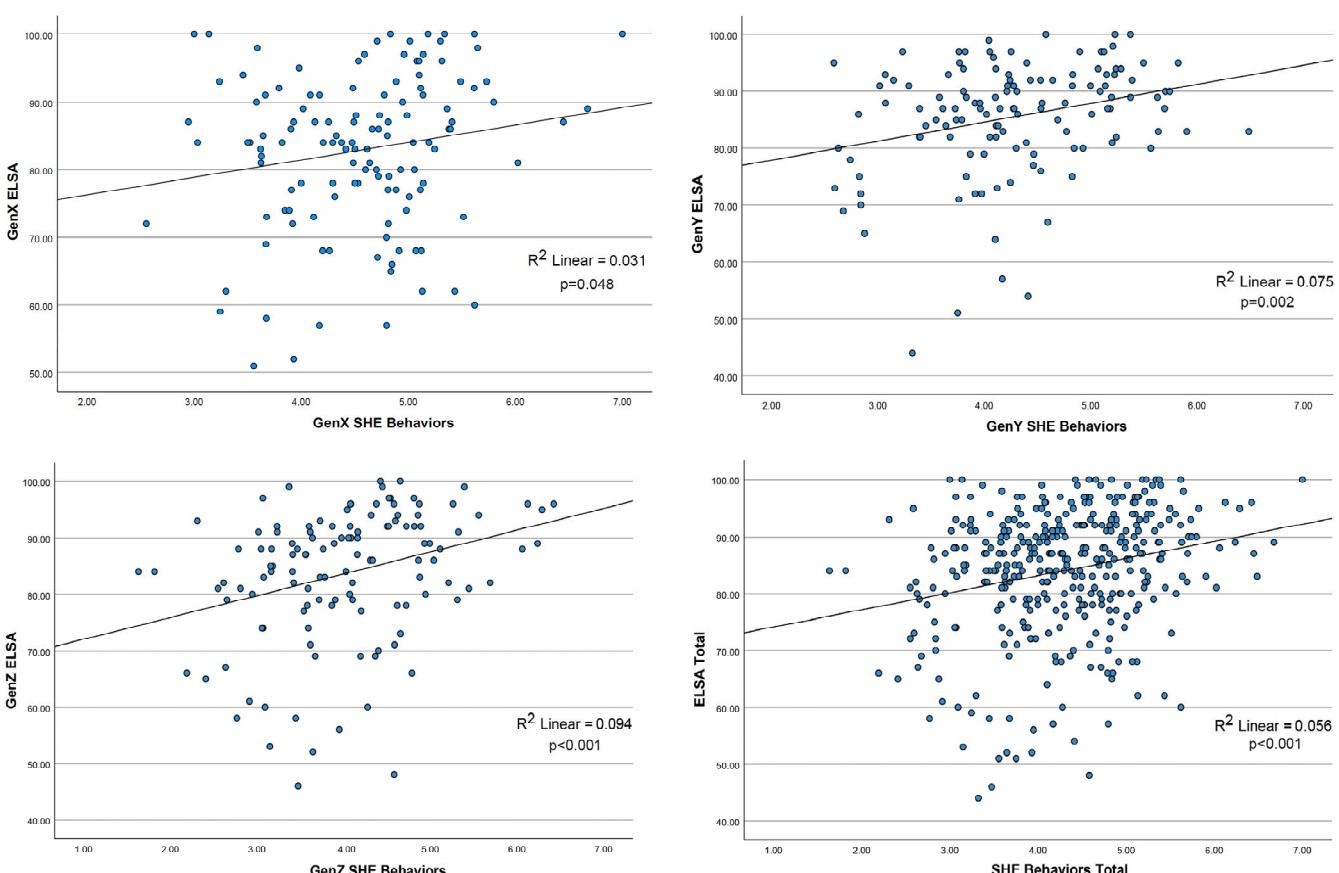

**Figure 1.** The correlation between the scores on the SHE Behavior Scale and the ELSA of the GenX, GenY, and GenZ participants. (According to linear regression results: GenX SHE Behaviors: 3.57 + GenX ELSA × 0.012; GenY SHE Behaviors: 2.34 + GenY ELSA × 0.023; GenZ SHE Behaviors: 1.99 + GenZ ELSA × 0.024; SHE Behaviors Total: 2.73 + ELSA Total × 0.019).

## 4. Discussion

The study involved 381 GenX, GenY and GenZ participants who shared the same ancestral background and assessed the relationship between SHE behaviors and environmental literacy. It was found that the total score of GenX participants on the SHE Behaviors

Scale was significantly higher ($p < 0.001$). This result implies that GenX participants may exhibit a greater adoption of SHE behaviors. In Turkey, as in the rest of the world, GenX members have been exposed to rapid changes and are a generation that constantly strives to keep up with these changes. The large proportion of GenX in the labor force affects the purchasing power of this generation. In addition, this situation may have provided easier access to foods produced with modern food production systems and more convenience and freedom in food choices [17]. GenX members are recognized in the business world as more competitive, hardworking, and individualistic [18]. Durukan and Gül [6] also found that today's GenX participants are aware that they are getting older and are therefore more conscious of their eating habits. In this context, in addition to generational characteristics, other socio-economic factors may also play a role in GenX's high scores on the SHE behavior scale. Moreover, in the factor analysis, it was determined that the "Quality Labels" ($p = 0.001$), "Meat Reduction" ($p = 0.011$) and "Low Fat" ($p = 0.026$) factor scores were similar in GenX and GenY. These factor scores were significantly higher compared to GenZ. The "Animal Welfare" factor score was significantly higher in GenX ($p = 0.006$). It can be asserted that GenX and GenY participants prioritize food quality and meat reduction and eating food with a low fat content. In addition, GenX participants were more concerned about "Animal Welfare". In the factors of "Avoiding Food Waste and Seasonal Foods" ($p < 0.001$) and "Local Food" ($p < 0.001$), it was observed that GenX participants scored significantly higher than GenY and GenZ participants. This suggests that GenX participants place a higher value on seasonal and local foods, showing greater awareness and consciousness in avoiding food waste. The results of this study were similar to those found in a study that examined the results of 2187 households in order to understand behaviors related to food waste in various generations [19]. It is known that individuals influenced by similar periods and societal factors develop different lifestyle habits, and their food preferences align with these influences. Upon analyzing the food consumption patterns of generations, it is observed that baby boomers and GenX are more conscientious about consuming organic and certified food products [20,21]. Moreover, GenX expresses a heightened concern about climate change compared to subsequent generations [22]. It was therefore important to discover that GenZ did not score highly on the SHE behaviors scale. It is valuable to investigate whether this is due to changing values, economic constraints, or a different understanding of sustainability.

GenX and GenY participants had significantly higher scores on the "Healthy and Balanced Nutrition" factor than GenZ participants ($p < 0.001$). In fact, in this study the majority of GenZ participants had one or both parents in either GenX or GenY. The fact that GenX and GenY scored significantly higher on the "Healthy and Balanced Nutrition" factor can be explained by the fact that it is not enough for GenX and GenY participants to transfer the behaviors they apply in terms of healthy and balanced nutrition to GenZ participants. In addition, GenZ participants are more likely to be influenced by their peers, as they are more open to internet technology and interpersonal communication [23]. In addition, GenY participants consumed significantly less fat than GenZ participants, and GenY participants are generally more active in their lives, which may have resulted from not wanting to gain weight.

There was no difference between the total score obtained from the ELSA for GenX, GenY and GenZ participants. In fact, although previous studies [24,25] have shown that GenZ is more sensitive to environmental health and environmental problems, this study did not find similar results. However, the "Environmental Consciousness" factor score was significantly lower in GenX participants than in GenY participants ($p = 0.007$). It is thought that the higher 'environmental consciousness' of GenX participants is probably influenced by the fact that their parents, the baby boomers, were more disciplined and more connected to nature.

The GenX participants exhibited positive correlation ($p = 0.048$), aligning with the conclusion that an increase in their SHE behaviors corresponds to an increase in their environmental literacy. This suggests potential synergy between environmentally con-

scious dietary choices and increased understanding of environmental issues in the GenX participants. Similarly, the positive correlation observed among GenY participants was not only significant but also more pronounced ($p = 0.002$), highlighting a stronger link between SHE behaviors and environmental literacy in this generation. The strong correlation in GenY suggests that participants who show a greater commitment to sustainable and healthy eating also tend to have a deeper understanding of environmental issues. The strongest correlation was observed in GenZ participants ($p < 0.001$), highlighting a particularly robust relationship between SHE behaviors and environmental literacy in the youngest generation. This finding underlines the critical role of sustainable dietary choices in shaping the environmental consciousness of the GenZ participants and aligns with their reputation for being more technologically savvy and interconnected. Overall, the positive correlations across all generations suggest a consistent relationship between SHE behaviors and environmental literacy.

It is observed that individuals in GenZ are more aware of sustainability, yet they lack SHE behaviors due to factors such as procurement, accessibility, and adoption [20,26]. In this study, despite higher environmental literacy in GenZ and GenY, the lag in SHE behaviors compared to GenX can be explained by similar factors. The development of healthy eating habits among older individuals further reinforces SHE behaviors [27]. The result that GenZ are aware of SHE behaviors but lack the ability to translate this awareness into behavior may offer intervention opportunities for future studies.

While it is theoretically acknowledged from a sociological standpoint that there exists a transfer of knowledge and experience between generations, it is also evident that each generation can cultivate distinct consumption habits based on their individual preferences and lifestyles [27]. In our study, significant correlation was observed with the next generation in terms of SHE behaviors; however, no relationship was identified between GenX and GenZ. This lack of association may be attributed to the prevalent nuclear family structure, particularly common in developing countries. With less frequent interaction with grandparents, the influence from this generation is expected to be minimal.

To the best of our knowledge, this study is the first study to evaluate sustainable and healthy eating behaviors and environmental literacy together in terms of generational differences, which may lead to future studies. Therefore, promoting and discussing the results of the study on social media platforms can provide valuable perspectives and information exchange [28]. The main limitation of this study is that the grandparents and grandchildren were not selected from the same gender (e.g., male–male or female–female– female). In addition, as the dynamics shaping each culture will be different, more studies examining similar issues in different cultures are needed.

## 5. Conclusions

In conclusion, it is evident that generational differences play a crucial role in shaping SHE behaviors and environmental consciousness. Particularly, the study suggests that GenZ, as potential future parents, should be given special consideration in the development of initiatives aimed at fostering consciousness and education on sustainable nutrition. Tailoring interventions to address the unique characteristics and preferences of each generation will likely contribute to more effective and impactful sustainable lifestyle initiatives.

**Author Contributions:** Conceptualization, N.Ö. (Neslihan Öner); Methodology, N.Ö. (Neslihan Öner), H.D. and A.B.; Formal analysis, H.D. and Y.Y.F.; Investigation, N.Ö. (Neslihan Öner), H.D., Y.Y.F. and N.Ö. (Nilüfer Özkan); Data curation, N.Ö. (Nilüfer Özkan); Writing—original draft, N.Ö. (Neslihan Öner), H.D., Y.Y.F., A.B. and N.Ö. (Nilüfer Özkan); Writing—review & editing, N.Ö. (Neslihan Öner), H.D., Y.Y.F. and A.B.; Project administration, N.Ö. (Neslihan Öner). All authors have read and agreed to the published version of the manuscript.

**Funding:** This research received no external funding.

**Institutional Review Board Statement:** The study procedure was performed according to Declaration of Helsinki. The study was approved by the Social and Human Sciences Ethics Committee of Erciyes University (Date 14 October 2022 and Approval No: 2022/04.08).

**Informed Consent Statement:** Participants were informed about the study and their written consent was obtained. Written consent was also obtained from underage participants' parents, who were also participants in the same study, were informed and gave consent for their children's participation.

**Data Availability Statement:** All data generated or analyzed during this study are included in this article. Further enquiries can be directed to the corresponding author.

**Conflicts of Interest:** The authors have no conflicts of interest to declare.

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
