# Peer review of "Sustainable and Healthy Eating Behaviors and Environmental Literacy of Generations X, Y and Z with the Same Ancestral Background: A Descriptive Cross-Sectional Study"

_sustainability, doi:10.3390/su16062497_

Round 1

Reviewer 1 Report

Comments and Suggestions for Authors

This study investigated the relationship between sustainable and healthy eating behaviors and environmental literacy of members of GenX, GenY and GenZ with the same ancestral background. This research is not only intriguing but also holds the potential to contribute novel insights to the field. However, after a careful review, I think there are still some minor issues to be addressed before acceptance

1. The repetitive annotation of abbreviations, such as SHE in line 97.

2. The standard deviation needs to be added to the data in Tables 2 and 3. In Table 2, the authors note that "Superscript letters indicate statistical difference between groups," yet the significance level of this test remains unspecified. Additionally, a thorough verification of the accuracy of the lowercase superscript letters in Table 2 is advised. It is imperative to verify the consistency between the results presented in lines 153-156 and the statistical significance test outcomes in Table 2.

3. Please confirm lines 87, 95, and 121.

4. It is recommended to consider modifying the structure of the table in order to enhance its aesthetics and standardization.

Author Response

Comments and Suggestions for Authors

(Reviewer 1)

Dear Reviewer,

My colleagues and I appreciate your valuable contributes for this research article. Thank you for evaluating our article.

1.     The repetitive annotation of abbreviations, such as SHE in line 97.

Author's Note to Reviewer:

We check throughout the article and corrected the repetitive annotation of abbreviation.

2.     The standard deviation needs to be added to the data in Tables 2 and 3. In Table 2, the authors note that "Superscript letters indicate statistical difference between groups," yet the significance level of this test remains unspecified. Additionally, a thorough verification of the accuracy of the lowercase superscript letters in Table 2 is advised. It is imperative to verify the consistency between the results presented in lines 153-156 and the statistical significance test outcomes in Table 2.

Author's Note to Reviewer:

“…The standard deviation needs to be added to the data in Tables 2 and 3.”

According to the characteristics of the data, the mean was presented with median and min-max (Table 2 and Table 3).

“…In Table 2, the authors note that "Superscript letters indicate statistical difference between groups," yet the significance level of this test remains unspecified. Additionally, a thorough verification of the accuracy of the lowercase superscript letters in Table 2 is advised.”

Superscript letters indicate statistical difference between groups (p < 0.05).

“…It is imperative to verify the consistency between the results presented in lines 153-156 and the statistical significance test outcomes in Table 2.”

Thank you for your attention. We corrected the statement: “… Body weight and BMI median values of the participants from all three generations were significantly different from each other.” Also, we checked the Table 2 and Table 3.

3.     “…Please confirm lines 87, 95, and 121.”

Author's Note to Reviewer:

Changed to [removed for single-blind peer review process].

4.     “… It is recommended to consider modifying the structure of the table in order to enhance its aesthetics and standardization.”

Author's Note to Reviewer:

Thank you for your attention. We corrected the table where aesthetics and standardization were required.

Reviewer 2 Report

Comments and Suggestions for Authors

Thank you for inviting me to review the paper - Sustainable and healthy eating behaviors and environmental literacy of the generations X, Y and Z with the same ancestral background: A descriptive cross-sectional study

some recommendations are as follow:

abstract - should follow journal style format - eg: "background", "methods",.... should be remove

why is it crucial to explore the relationship between sustainable eating behaviors and environmental literacy across generations with common ancestry - not that convincing starting sentence for the abstract - can be improve

introduction

should expand more on how sustainable eating behaviors and environmental literacy are related, since this is the core theoretical framework for the current study

simply providing the definition is not enough

are there hypothesis? specific research questions?

tables should be in the proper style formatting

P should be small caps

statistical results seems adequate - however would suggest to use regression to provide further predictive measures and not just correlations between sustainable eating behaviors and environmental literacy

would suggest to provide the explain power for the ANOVAs

how about post-hoc analysis?

Author Response

Comments and Suggestions for Authors

(Reviewer 2)

Dear Reviewer,

My colleagues and I appreciate your valuable contributes for this research article. Thank you for evaluating our article.

1.     “… abstract - should follow journal style format - eg: "background", "methods”.... should be remove.”

Author's Note to Reviewer:

Thank you for your attention. We corrected “The Abstract” section like journal style format.

2.     “… why is it crucial to explore the relationship between sustainable eating behaviors and environmental literacy across generations with common ancestry - not that convincing starting sentence for the abstract - can be improve”

Author's Note to Reviewer:

 Thank you for your valuable contribution. We just expressed why it’s crucial to explore this relationship within the word limit of abstract.

“…As social culture and structure evolve, changes occur in individuals' eating habits and environmental awareness. This study assesses the relationship between sustainable eating behaviors and environmental literacy across generations (Gens) from the same ancestry.”

3.     “… should expand more on how sustainable eating behaviors and environmental literacy are related, since this is the core theoretical framework for the current study … are there hypothesis? specific research questions?”

Author's Note to Reviewer:

Thank you for your valuable contributions. We have added research questions to the study. Thus, we think that “The introduction” section is enriched in terms of the importance of the subject examined in the study.

“…The main research questions were:

RQ1. Is there a relationship between sustainable and healthy eating behaviors and environmental literacy of different generations?

RQ2. Which factors affect the sustainable and healthy eating behaviors and environmental literacy of different generations?”

4.     “… tables should be in the proper style formatting P should be small caps”

Author's Note to Reviewer:

Corrected.

5.     “…statistical results seems adequate - however would suggest to use regression to provide further predictive measures and not just correlations between sustainable eating behaviors and environmental literacy”

Author's Note to Reviewer:

Done. “… According to linear regression results; GenX SHE Behaviors: 3.57 + GenX ELSA x 0.012, GenY SHE Behaviors: 2.34 + GenY ELSA x 0.023, GenZ SHE Behaviors: 1.99 + GenZ ELSA x 0.024, SHE Behaviors Total: 2.73 + ELSA Total x 0.019.”

6.     “…would suggest to provide the explain power for the ANOVAs”

Author's Note to Reviewer:

“…The actual power calculated by post hoc power analysis was 97%.”

7.     “…how about post-hoc analysis?”

Author's Note to Reviewer:

Corrected for Table 2 and Table 3.

Reviewer 3 Report

Comments and Suggestions for Authors

I revised the paper entitled: Sustainable and healthy eating behaviors and environmental 2 literacy of the generations X, Y and Z with the same ancestral 3 background: A descriptive cross-sectional study.

Please find here my specific comments that needs to be addressed to improve the overall clarity of the paper.

Line 9-23: I suggest you to follow the journal formatting guidelines, therefore the abstract should be a maximum of 250 words and not specifying the introduction, material and methods etc… Please do write it as a whole text without the divisions.

Lines 26-34:  It would be beneficial to briefly mention how these sociocultural environments specifically influence dietary habits, to directly link this background to the study's focus on sustainable eating and environmental literacy

Lines 26-34: The statement effectively sets a broad historical context for generational differences in dietary habits. However, it would be beneficial to briefly mention how these sociocultural environments specifically influence dietary habits, to directly link this background to the study's focus on sustainable eating and environmental literacy

Lines 73-76: Consider specifying the criteria used to determine "the same ancestral background" among participants. This clarification can help readers understand how this factor might influence the study's findings.

Lines 86-90: The age range definitions for Generations X, Y, and Z are clear. Yet, a comment about considering regional or cultural differences in these generational boundaries might be useful, as they can vary globally.

General comment for the results section: All the tables presented in the Results section do not adhere to the formatting guidelines of the journal. Specifically, the organization and presentation of data within these tables may not meet the required standards for clarity, consistency, or accessibility. As such, it is recommended that the authors revisit these tables, ensuring that they align with the journal's guidelines for table formatting. This may involve adjustments to the layout, labelling, scaling, or other aspects of data presentation to enhance readability and comprehension. By reorganizing the data more effectively, the authors can facilitate a better understanding of the study's findings, allowing readers to easily grasp the key results and their implications.

Lines 190-194: The observation that GenX scored significantly higher on the SHE Behaviors Scale is compelling. However, it would be enriching to discuss potential factors influencing this higher score. Is it solely due to generational characteristics, or could other socio-economic or cultural factors play a role?

Lines 195-197: The claim that GenX strives to keep up with rapid changes and therefore exhibits more sustainable eating habits could benefit from further elaboration. It may be useful to draw on additional literature to support the link between these generational characteristics and SHE behaviors.

Lines 200-208: While the significant scoring in specific factors like "Quality Labels" and "Local Food" by GenX is interesting, it could be valuable to explore why younger generations, particularly GenZ, might not score as highly. Is this a reflection of changing values, economic constraints, or perhaps a different understanding of sustainability?

Lines 217-222: The point about GenX and GenY's higher scores in "Healthy and Balanced Nutrition" raises questions about the transmission of values and habits across generations. It would be insightful to discuss strategies or mechanisms through which these healthier behaviors could be more effectively passed down to GenZ.

Lines 236-251: The correlation between SHE behaviors and environmental literacy is a crucial finding. Discussing the implications of these correlations for environmental and nutritional education programs could offer valuable insights into how different generations could be targeted or engaged.

Lines 252-256: Highlighting GenZ's awareness yet lack of SHE behaviors presents an opportunity for intervention. Expanding on the barriers to adopting SHE behaviors among younger generations could guide future research and program development.

I highly recommend you to add a paragraph which addresses the study limitations. Including a paragraph on limitations would not only enhance the credibility of the study but also help in framing the findings within the context of these constraints.

Additional points:

To enhance the impact of their study and engage a wider audience, the authors could consider the following suggestions:

Use Social Media Platforms: The authors should leverage popular social media platforms such as Twitter, Facebook, Instagram, and LinkedIn to share their findings. They can create engaging posts that highlight the specific factors of sustainable eating behavior, like "Quality Labels," "Animal Welfare," and "Meat Reduction," and compare the differences between Generation X and Generation Z. Visual content such as infographics, charts, and short videos can help convey key points effectively.

Report Actual Profiles: Sharing real-life examples of individuals or organizations that exemplify the sustainable eating behaviors observed in Generation X can add credibility and relevance to the study. They could highlight profiles of influencers, activists, or companies that prioritize sustainability in their food choices and practices. By showcasing these examples, the authors can demonstrate how certain generational attitudes and behaviors are already shaping the food landscape.

Reference Recent Published Papers: Incorporating references to recent papers or studies that corroborate their findings can strengthen the validity of their research. The authors can cite scholarly articles, reports, or news pieces that explore similar themes or trends in sustainable eating behaviors across different generations. By providing additional context and supporting evidence from other sources, they can enrich their discussion and stimulate further interest in the topic.

Engage with Online Communities: Engaging with online communities focused on sustainability, nutrition, and environmental awareness can help the authors reach a broader audience and spark meaningful discussions. They can participate in relevant forums, groups, and discussions on platforms like Reddit or specialized online forums dedicated to food sustainability and healthy eating. By sharing insights from their study and inviting feedback from community members, they can foster dialogue and exchange valuable perspectives.

By implementing these strategies, the authors can effectively disseminate their research findings, stimulate interest and discussion, and contribute to raising awareness about sustainable eating behaviors and environmental literacy across different generations. Please see and cite: 10.3390/ani13223503.

Author Response

Comments and Suggestions for Authors

(Reviewer 3)

Dear Reviewer,

My colleagues and I appreciate your valuable contributes for this research article. Thank you for evaluating our article.

1.     “…Line 9-23: I suggest you to follow the journal formatting guidelines, therefore the abstract should be a maximum of 250 words and not specifying the introduction, material and methods etc… Please do write it as a whole text without the divisions.”

Author's Note to Reviewer:

Thank you for your attention. We corrected “The abstract” section.

“Abstract: As social culture and structure evolve, changes occur in individuals' eating habits and environmental awareness. This study assesses the relationship between sustainable eating behaviors and environmental literacy across generations (Gens) from the same ancestry. Sustainable and Healthy Eating (SHE) Behavior Scale and the Environmental Literacy Scale for Adults (ELSA) was administered to 381 individuals across all generations. Self-reported anthropometric data were collected. The total score of the SHE Behaviors Scale of the participants from all three generations were significantly different from each other. "Quality Labels", "Meat Reduction" and "Low Fat" factor scores were similar in GenX and GenY. These factor scores were significantly lower in GenZ. “Animal Welfare” factor score was significantly higher in GenX. "Avoiding Food Waste and Seasonal Foods" and "Local Food" factor scores were significantly higher in GenX than in GenY and GenZ. "Healthy and Balanced Diet" factor score was significantly lower in GenZ. There was no difference between the total ELSA scores. "Environmental Consciousness" factor score was significantly lower in GenX than in GenY. Generational disparities strongly influence perspectives on sustainable and healthy eating. Focused initiatives are essential to educate future parents, who play a pivotal role in shaping the next generation, about sustainable nutrition.”

Word count: 199

2.     “…Lines 26-34:  It would be beneficial to briefly mention how these sociocultural environments specifically influence dietary habits, to directly link this background to the study's focus on sustainable eating and environmental literacy”

3.     “…The statement effectively sets a broad historical context for generational differences in dietary habits. However, it would be beneficial to briefly mention how these sociocultural environments specifically influence dietary habits, to directly link this background to the study's focus on sustainable eating and environmental literacy.”

Author's Note to Reviewer:

Thank you for your valuable contribution. We hope that it will be more fluent for the reader if it is emphasized as you have stated.

“… In a study conducted by Zeren et al. [4] it was observed that individuals who are members of different generations have different nutritional behaviors; Generations X and Y prefer a healthier nutritional perspective; however, Generation Z exhibits a hedonistic approach to eating habits. Although the nutritional knowledge of Generation Z is higher than other generations, it is stated that their interest in nutrition is more related to their physical appearance. Different perceptions about nutrition are likely to create different eating habits in different generations.”

4.     “…Lines 73-76: Consider specifying the criteria used to determine "the same ancestral background" among participants. This clarification can help readers understand how this factor might influence the study's findings.”

Author's Note to Reviewer:

Specified the criteria of the same ancestry background:

“… The same ancestral background of the participants was determined as grandparent-parent-grandchild.”

5.     “…Lines 86-90: The age range definitions for Generations X, Y, and Z are clear. Yet, a comment about considering regional or cultural differences in these generational boundaries might be useful, as they can vary globally.”

Author's Note to Reviewer:

The requested comment was added.

“… Considering that there may be regional or cultural differences in the classification of generations in this study, the age ranges of a study conducted in [5] [removed for single-blind peer review process] were preferred; individuals representing GenX were defined as those born between 1965-1979, individuals representing GenY were defined as those born between 1980-1999, and individuals representing GenZ were defined as those born in 2000 and later.”

6.     “… Lines 190-194: The observation that GenX scored significantly higher on the SHE Behaviors Scale is compelling. However, it would be enriching to discuss potential factors influencing this higher score. Is it solely due to generational characteristics, or could other socio-economic or cultural factors play a role?”

Author's Note to Reviewer:

We discussed this situation also improved with a new comment.  

“…Durukan and Gül [6] also found that today's GenX participants are aware that they are getting older and are therefore more conscious of their eating habits. In this context, in addition to generational characteristics, other socio-economic factors may also play a role in GenX's high scores on the SHE behaviors scale.”

7.     “…Lines 195-197: The claim that GenX strives to keep up with rapid changes and therefore exhibits more sustainable eating habits could benefit from further elaboration. It may be useful to draw on additional literature to support the link between these generational characteristics and SHE behaviors.”

Author's Note to Reviewer:

8.     “…Lines 200-208: While the significant scoring in specific factors like "Quality Labels" and "Local Food" by GenX is interesting, it could be valuable to explore why younger generations, particularly GenZ, might not score as highly. Is this a reflection of changing values, economic constraints, or perhaps a different understanding of sustainability?”

Author's Note to Reviewer:

9.     “… Lines 217-222: The point about GenX and GenY's higher scores in "Healthy and Balanced Nutrition" raises questions about the transmission of values and habits across generations. It would be insightful to discuss strategies or mechanisms through which these healthier behaviors could be more effectively passed down to GenZ.”

Author's Note to Reviewer:

10.  “… Lines 236-251: The correlation between SHE behaviors and environmental literacy is a crucial finding. Discussing the implications of these correlations for environmental and nutritional education programs could offer valuable insights into how different generations could be targeted or engaged.”

Author's Note to Reviewer:

11.  “…Lines 252-256: Highlighting GenZ's awareness yet lack of SHE behaviors presents an opportunity for intervention. Expanding on the barriers to adopting SHE behaviors among younger generations could guide future research and program development.”

Author's Note to Reviewer:

12.  “…I highly recommend you to add a paragraph which addresses the study limitations. Including a paragraph on limitations would not only enhance the credibility of the study but also help in framing the findings within the context of these constraints.”

Author's Note to Reviewer:

Added.

“…The main limitation of this study is that the grandparents-grandchildren were not selected from the same gender (e.g. male-male or female-female-female). In addition, as the dynamics shaping each culture will be different, more studies examining similar issues in different cultures are needed.”

13.  “…By implementing these strategies, the authors can effectively disseminate their research findings, stimulate interest and discussion, and contribute to raising awareness about sustainable eating behaviors and environmental literacy across different generations. Please see and cite: 10.3390/ani13223503.”

Author's Note to Reviewer:

“… To the best of our knowledge, this study is the first study to evaluate sustainable and healthy eating behaviors and environmental literacy together in terms of generational differences, which may lead to future studies. Therefore, promoting and discussing the results of the study on social media platforms can provide valuable perspectives and information exchange [27].”

27.    Muca, E.; Buonaiuto, G.; Lamanna, M.; Silvestrelli, S.; Ghiaccio, F.; Federiconi, A.; ... & Cavallini, D. Reaching a wider audience: Instagram’s role in dairy cow nutrition education and engagement. Animals. 2023, 13(22), 3503.

Reviewer 4 Report

Comments and Suggestions for Authors

Thank you for allowing me to review this manuscript.

The study aims to evaluate the relationship between sustainable eating behaviors and environmental literacy across generations X, Y and Z with common ancestry.

The introduction must be expanded and improved. This section is setting the general background for the research and also contains the literature review.

The studies covered are relevant for the research, yet the literature review must be richer and more diverse. Numerous previous recent studies are related to the topic of this research and could be reviewed, such as: https://doi.org/10.3389/fenvs.2023.1096183

The data and methods are adequately described. Please consider moving Table 1 to Annexes. It is too extensive occupying five pages from the main manuscript. Additionally, comprehensively explain the information presented by this table.

Please consider including a graphical representation of the results to emphasize the understanding of the main differences between categories included in the study.

The results are adequately discussed in the conclusions. Nevertheless, please present and discuss several implications of these findings. How markets can adapt to such behaviors? What are the perspectives for the business environment? Are there ecological implications that derive from it?

Please overview the entire manuscript for typos and styling.

Author Response

Comments and Suggestions for Authors

(Reviewer 4)

Dear Reviewer,

My colleagues and I appreciate your valuable contributes for this research article. Thank you for evaluating our article.

1.     “…The introduction must be expanded and improved. This section is setting the general background for the research and also contains the literature review.”

Author's Note to Reviewer:

Thank you for your valuable contribution. We have added research questions to the study. Thus, we think that “The introduction” section is enriched in terms of the importance of the subject examined in the study.

“…The main research questions were:

RQ1. Is there a relationship between sustainable and healthy eating behaviors and environmental literacy of different generations?

RQ2. Which factors affect the sustainable and healthy eating behaviors and environmental literacy of different generations?”

2.     “…The studies covered are relevant for the research, yet the literature review must be richer and more diverse. Numerous previous recent studies are related to the topic of this research and could be reviewed, such as: https://doi.org/10.3389/fenvs.2023.1096183.”

Author's Note to Reviewer:

Thank you for your valuable contribution. We have read the work and added it to the reference.

“… GenZ, on the other hand, is a generation that is more likely to be more conscious about environmental health [8], …”

With this emphasis, the results of our study have been supported and extended. 

3.     “… The data and methods are adequately described. Please consider moving Table 1 to Annexes. It is too extensive occupying five pages from the main manuscript. Additionally, comprehensively explain the information presented by this table.”

Author's Note to Reviewer:

After all revisions were completed, we could moving Table 1 to Annex.

4.     “…Please consider including a graphical representation of the results to emphasize the understanding of the main differences between categories included in the study.”

Author's Note to Reviewer:

5.     “…The results are adequately discussed in the conclusions. Nevertheless, please present and discuss several implications of these findings. How markets can adapt to such behaviors? What are the perspectives for the business environment? Are there ecological implications that derive from it?”

Author's Note to Reviewer:

6.     “…Please overview the entire manuscript for typos and styling.”

Author's Note to Reviewer:

The manuscript was checked for typos and styling. Corrected.

Round 2

Reviewer 2 Report

Comments and Suggestions for Authors

After going over the revisions made by the author/s the current version of the paper seems adequate for acceptance.

However, would recommend to provide some practical implications, what now? after the findings.

Author Response

Dear Reviewer, 

Thank you very much for evaluating our article and providing many valuable contributions. My colleagues and I have been working on sustainable nutrition for the last three years and we have a funded project that we are currently working on. The common goal of this study and all our other studies is to investigate what can be done to popularize sustainable nutrition and protect the health of the planet and to take action on this issue. An important finding of this study, that GenZ is aware of SHE behaviors but lacks the ability to transform this awareness into behavior, may offer intervention opportunities for future studies.  

Regards,

Reviewer 3 Report

Comments and Suggestions for Authors

The authors have adressed all the comments, in my opinion the paper is suitable for pubblication.

Author Response

Dear Reviewer, 

Thank you very much for evaluating our article and providing many valuable contributions.

Regards,

Reviewer 4 Report

Comments and Suggestions for Authors

The authors addressed the comments and suggestions for improving the manuscript. 

Author Response

Dear Reviewer, 

Thank you very much for evaluating our article and providing many valuable contributions. Your valuable comments have made our work better. In particular, your suggestion that we emphasize the opportunities that the findings of this study offer for future studies will be of interest to other researchers.

Regards,
